# Adolescent Depression from a Developmental Perspective: The Importance of Recognizing Developmental Distress in Depressed Adolescents

**DOI:** 10.3390/ijerph192316029

**Published:** 2022-11-30

**Authors:** Christopher Rikard-Bell, Caroline Hunt, Claire McAulay, Phillipa Hay, Arshia Morad, Michelle Cunich, Stephen Touyz

**Affiliations:** 1School of Psychology, Faculty of Science, University of Sydney, Sydney, NSW 2006, Australia; 2Faculty of Medicine, University of Western Sydney, Penrith, NSW 2751, Australia; 3School of Psychology and Inside Out Institute, University of Sydney, Sydney, NSW 2006, Australia

**Keywords:** adolescent, depression, dysphoria, developmental, self-perception

## Abstract

Objective: To make the case that developmental distress needs to be assessed when evaluating adolescent depression. Methods: Reviews of relevant papers relating to adolescent depression. Results: Adolescent depression is a common and costly health condition, confounded by a lack of consensus among health professionals regarding evidence-based approaches regarding treatments. Little attention has been paid to the contribution of developmental distress. Conclusion: The current adult-like model of adolescent depression fails to advance the understanding of adolescent depression. A systematic evidence-based approach to identifying developmental self-perception distress in depressed adolescents could provide important advances in treatment to improve short-term and longer-term mental health outcomes. This paper proposes the creation of a psychometric tool to systematically measure developmental self-perception distress in adolescents with depression.

## 1. Introduction

Adolescent depression and suicide are particularly important mental health concerns across all societies, with the additional worry that depression in adolescents potentially foreshadows a life-long battle with mental illness [1,2,3]. Adolescence heralds a rapid increase in mental disorders, including depression, and so there are excellent opportunities for interventions to be implemented during adolescence regarding the treatment and prevention of depression and other mental disorders [3,4,5]). Current approaches to diagnosis and treatment of adolescent depression are largely based upon adult depressive criteria, with an emphasis on treating the presenting symptoms, which we argue is too restrictive and leads to a confusing lack of clarity or uniformity of method regarding the various treatment options [3,6,7,8,9,10,11]. A systematic evidence-based approach to identifying developmental self-perception distress in depressed adolescents could provide important advances in treatment to improve short-term and longer-term mental health outcomes. The purpose of this paper is to provide an accessible succinct developmental paradigm, so that additional approaches to treatment can be applied utilizing developmental concepts. We propose a novel approach to assessment, and we also argue for the creation of a psychometric tool to systematically measure developmental self-perception distress to assist and guide the treatment of adolescents with depression.

## 2. The Problem

Adolescent depression is a particularly important mental health disorder, and its recognition and treatment have important short and long-term mental health implications. Adolescence is a critical time for mental well-being, as almost 75% of adult mental illnesses have origins during childhood and youth [1,2,3,12,13]. Adolescent depression is common, and almost 70% of adolescents who have a depressive disorder will have a recurrence within five years, with a four-fold risk of experiencing depression in adulthood [4,14,15]. Suicide is the second most common cause of death in this age group, with the rate of depression and suicide worsening in industrialized countries [16,17,18]. Any substantial improvements in the treatment of adolescent depression would clearly be welcomed. In adolescents, it is worth noting that bipolar depression (as part of bipolar affective disorder) is relatively rare, whilst non-bipolar adolescent depression is common and so most of this manuscript refers to non-bipolar adolescent depression [19,20,21].

## 3. The Gap

The gaps that we have identified for adolescent depression involve treatment confusion, the poor conceptualization of the disorder and the need for differentiation from adult depression due to a lack of clarity regarding the transition from childhood disorders to adult disorders.

The treatment outcomes for adolescent depression are only moderately effective and so remain confusing and stubbornly disappointing [12,16,22,23]. Accurately identifying appropriate treatments for depressed adolescents is particularly challenging and important. However even when administered in a timely way, treatments have not resulted in the expected reduction in mortality and morbidity, despite increased specialty adolescent mental health providers, increased use of prescription medications, and higher rates of hospitalizations [16]. In their Cochrane review, it was concluded there is insufficient empirical evidence to inform a coherent approach to the treatment of adolescent depression as the relative effectiveness of psychological interventions, antidepressants, or a combination of these treatments could not yet be established and there was no statistical difference between various types of psychological and antidepressant treatments [24]. Further, it was unclear whether there was any treatment that had any impact on suicide rates. Nevertheless, the review noted that untreated depression is likely to continue into adulthood. Another Cochrane review examined evidence-based psychological prevention programs for adolescent depression [25]. However, this review concluded that there was inadequate evidence to support the implementation of interventions, even with recognized evidence-based psychological prevention programs. A further review found that in adolescents with moderate to severe depression, fluoxetine, either alone or with cognitive behavioral therapy (CBT), was better than placebo [26]; however, the treatment responses for antidepressants and psychological treatments were significantly less effective when compared with the results seen in adults.

In a literature review for the treatment of adolescent unipolar major depression, it was found that the various treatment approaches to depressed adolescents in the Brief Psychological Intervention of the IMPACT study were better than placebo [23]. Interestingly, those non-specific psychological treatments (comprising a collaborative approach and psychoeducation, selected behavioral activation techniques and recovery support methods) appeared to be as effective as specific psychological treatments (CBT and interpersonal therapy) [27]. In addition, the results for resilience and school prevention programs were short-lived, but there was some support for exercise, psychodynamic therapy and family therapy [22]. Adolescents who have a history of maltreatment responded better to a relational form of psychotherapy.

Studies regarding the efficacy of antidepressants in adolescent depression have been disappointing. It has long been recognized that, with few exceptions, antidepressants are arguably not better than placebo and are not recommended for adolescents with depression [28]. The most effective antidepressant was fluoxetine, with some evidence for sertraline and escitalopram [23]. In a 2016 meta-analysis of 30 controlled studies comparing selective serotonin reuptake inhibitors (SSRI) and placebo, the researchers found unconvincing overall evidence for antidepressants [29]. There were only two studies that showed improvement with fluoxetine compared to placebo [30,31]). In a more recent review, it was reconfirmed that fluoxetine alone and in combination with CBT was effective, with more response demonstrated with severe endogenous depression (bipolar-like depression) than with milder forms of depression [22]. Therefore, the evidence for antidepressant medication as the optimal approach to the treatment of the majority of adolescents with depression remains unclear. With adolescent depression common in primary care settings, the relevance of the rule of diminishing halves in primary settings may be important and mental disorders in the community were often confounded by poor recognition rates, poor compliance to treatment, and poor response to treatment [32].

Adolescent depression is most likely different from adult depression, and the term adolescent depression is often an umbrella term that encompasses a range of conditions, including anxiety, substance use, obsessive-compulsive disorder (OCD), and eating disorders. Although it is widely acknowledged that adolescents may present differently clinically from adults, such as description-atypical depression, a significant contributor to the difficulty in treating adolescents with depression is that there is still no consensus regarding the essential clinical features. Differing sets of diagnostic criteria have been proposed with essential clinical features, such as dysphoria and general impairment in functioning, and associated features, such as low self-esteem, guilt, and pessimism [2,3,33,34]. Adolescent depression is often associated with inchoate dysphoric emotions with depressive, anxious, and behavioral symptoms [2,35]. It was found that adolescents have high rates of dysphoria in the form of misery and self-doubt that go largely undetected, as identified initially in the Isle of Wight Study [22,36,37]. Dysphoria describes a general state of dissatisfaction and unease associated with emergent anxiety and depressive symptoms. In order to achieve uniformity, without empirical support, operationalized adult depression criteria were surprisingly adopted for adolescents and children [3,6,10]. In what appears to be an acknowledgment of perhaps some differences developmentally, irritability, as an additional element, has been added for depression in children and adolescents in the Diagnostic and Statistical Manual of Mental Disorders Fifth Edition (DSM-5) as a presenting depressive symptom [38].

The mode of transition of diagnoses from childhood and adolescence to adulthood is still lacking clarity for many disorders, and yet diagnostic transitions are central to understanding how adolescent disorders relate to adult disorders [2,39]. Mentally unwell adolescents may be predisposed to later developing different types of adult mental disorders (heterotypic), or they may have an early form of an adult disorder (homotypic). It is appropriate that some adolescent-onset psychiatric disorders should be regarded as having homotypic transitions as they signal the early onset of lifelong adult psychiatric disorders, such as schizophrenia, bipolar affective disorder, and attention deficit hyperactivity disorder [2,35,37]. Bipolar depression in adolescents is important to recognize and is typically homotypic [19]. Even bipolar depression may present with different symptoms as some of the symptoms of early presentation of bipolar depression may not necessarily be easily recognized and may be appropriately explained by the diagnostic clinical staging proposal [40].

Adolescents diagnosed with non-bipolar depression present an increased risk for depression later in life, yet how many depressed adolescents progress to depressed adults is still unclear. Counterintuitively, depressed adolescents may not simply be suffering from an early form of adult-type depression. For example, researchers found much greater heterotypic transitions for emotional disorders and concluded that a disorder in adolescence was a risk for a range of heterotypic psychiatric disorders [41]. In another systematic review, researchers also confirmed that non-bipolar depressed adolescents have an increased risk not only for adult depressive disorders but also for anxiety disorders [39]. This document does not attempt to fully deal with the complex issues concerning developmental continuity and discontinuity of disorders. However, it should be noted that adolescent depression may not simply be an early form of adult depression, which would be an example of a post hoc ergo propter hoc fallacy [2,39,42].

Risk factors for this age group are complex and can be drawn from well-known biological, social, and psychological determinants that cover early adverse life experiences as well neurobiological disadvantages [2,3,22]. Some researchers have attempted to look beyond recognized risk factors to identify particular vulnerability factors, and so the search for early psychopathological factors is revealing interesting findings. Researchers have recommended it was necessary to reformulate the clinical phenotype of adolescent depression, as they argued there was growing evidence for an as yet unidentified latent distress-psychopathology trait or common psychological factor ‘p’ factor [23]. Potentially as yet undiscovered factors may contribute greatly to the classification and understanding of psychiatric symptoms as is being empirically studied by the HiTOP group [43]. The ‘p’ factor is yet to be clarified as to whether it is biological or non-biological in nature, and it could, in fact, possibly represent a latent genetic factor or possibly an underlying undifferentiated level of developmental distress in non-bipolar depressed adolescents. The HiTOP group notably is not without its critics as it does not inform treatment [44]. With regard to the brain-derived neuropathic factor (BDNF) the findings were the opposite of adult studies, which suggested different underlying mechanisms of action between antidepressants between adults and adolescents [22] The truth in understanding adolescent depression is that there is most likely a complex mix of biological and non-biological vulnerabilities that interact with adversity resulting in both continuity and discontinuity of symptoms and disorders.

## 4. A New Approach to Adolescent Depression

Moving away from the uncertainties of diagnosis classification and labels as the guides for treatment, we argue for refocusing on the developing psychology of the adolescent as a potential way forward for improving treatment in depressed adolescents. Even though risk factors are important for shaping the adolescent’s developmental experiences, it has been difficult to intervene and provide treatment for all biological and non-biological risk factors, and so our approach is to target risk factors that impact the adolescent’s self-perception. The risk factors we recommend targeting are those self-perceptions emanating from developmental experiences, including, for example, early family instability and bullying, which impact self-security, self-esteem, and self-image [13,45,46,47,48]. Mental well-being is regarded as a combination of nature and nurture, and so if nature and nurture are both important, then adolescents’ developmental experiences may be important in how many adolescents develop adolescent depression. It is argued here that the high level of adolescent undifferentiated emotional distress in depressed adolescents could be regarded as a form of developmental distress resulting in dysphoria. Although there may be differing approaches to understanding adolescent depression, there is strong evidence that depressed adolescents are struggling with negative self-perception [12,45,47]. Self-perception is important in depressed adolescents when considering the high self-reported rates of internal insecurity, poor coping perceptions, poor self-efficacy, low self-worth, poor self-esteem, and disturbed self-image [13,45,46,47,48,49].Adding further weight to self-perceptions importance is the strong correlation between the severity of self-image disturbance and the severity of adolescent depression [50,51,52]. While there are other biological and psychological models used to understand adolescent depression, the developmental distress seen in adolescents with depression may also explain the range of undifferentiated affective symptoms. Due to the conceptual roadblocks in the field, we sought to suggest a developmental distress paradigm of adolescent depression that considers developmental disruptions to the developing self-perception as key to identifying a viable target of intervention. The challenge will be to objectively quantify and objectify items for the self-perception paradigm. Our research group plans to develop a testable developmental model and validate a developmental psychometric instrument, utilizing well-established theoretical constructs, including attachment theory and life stage theory [53,54]. It is anticipated that preschool, primary school, and high school are obviously separated time frames that correlate with important developmental stages to inform the adolescent’s self-perception and possibly identify if there are ongoing developmental distresses. For example, adolescents who have had severe adverse early attachment disruption may have self-perception distress regarding fears of abandonment; further, if adolescents experienced previous severe bullying in primary school, they may have self-perception distress regarding fear of not coping or of being inadequate; or in high-school adolescents may have severe self-perception distress relating to body image or fear of being judged negatively. Once an adolescent has been diagnosed with non-bipolar adolescent depression, the developmental distress will need to be quantified to guide the clinician to a formulation from which to tailor treatment with attention to one or more of these stages of development of self-perception. Treatment could potentially use the developmental distress formulation and the evolution of one’s self-perception in the psychoeducation phase of treatment, followed by ways to explore the distress underneath the thwarted self and ways to rectify and correct this.

In brief, the self-perception paradigm we propose borrows from various developmental theorists as well as schema therapy, CBT, and narrative therapy, and combines and summarizes psychosocial development based upon a concept termed Developmental Sensitivity Theory (DST). Sensitivity in this context is used in a similar way to how imprinting is observed in the attachment-sensitive behavior of many animals, and perhaps most famously, goslings seen imprinting upon and following the early researcher Konrad Lorenz [55]. The DST argues that the critical parts of human psychological development can be condensed into the three most important sensitive parts of childhood development (correlating with the three cognitive maturational steps), which have similar qualities to imprinting, as the sensitive construct persists beyond the phase of the child’s development in the form of an internalized construct, which by adolescence becomes an internalized narrative. The DST suggests that there are three critical key areas of biopsychosocial developmental difficulties resulting in disordered internalized narratives that include: disorder of secure-base, disorder of competency, and disorder of self-image, which broadly correlate to preschool-age, primary school-age, and high school-age. An adolescent who experienced significant disturbance or disruption of any or all of these sensitive developmental periods will have enduring disordered internal narratives associated with the disturbed developmental period. The three core developmental constructs are related first to the core feeling of being unsafe, secondly to the core feeling of being unable to cope, and thirdly to the core feeling of being socially rejected. Whereas schema therapy has many potential schemas, DST proposes three core potential pathological developmental narratives. DST is different from CBT as pathological narratives can coexist or accompany the variety of CBT cognitive distortions.

Inevitably, there will be criticisms of this approach to understanding adolescent depression. For example, this developmental approach may arguably not account well for underlying neurodevelopmental conditions, such as intellectual disability, attention deficit hyperactivity disorder (ADHD), autistic spectrum disorder (ASD), obsessionality, and OCD. However, we argue that the developmental self-perception distress approach may still reflect the impact of these neurodevelopmental influences on psychological development. For example, children with ADHD or ASD may be predisposed to experiencing bullying or marginalization and feel self-perception distress about being able to cope; or extremely obsessional adolescents may struggle with self-perception distress about self-image.

Therefore, we argue adolescent depression is of such importance that it is imperative innovative approaches to assessment and treatment be trialed. We believe that a greater focus on assessing developmental distress in depressed adolescents will open up a better understanding of adolescent depression and also greater treatment opportunities. To achieve this goal, the creation of a systematic psychological developmental tool will allow another layer of clinical judgment to quantify and understand adolescent depression by identifying areas of developmental self-perception distress to better assist and guide approaches to treatment.

## 5. Conclusions

The current adult-like model of adolescent depression fails to advance the understanding and, therefore, treatment of adolescent depression. A systematic evidence-based approach to identifying developmental self-perception distress in depressed adolescents could provide important advances in treatment to improve short-term and longer-term mental health outcomes.

## Data Availability

Not applicable.

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
