# Peer review of "Adolescent Depression from a Developmental Perspective: The Importance of Recognizing Developmental Distress in Depressed Adolescents"

_ijerph, 2022, doi:10.3390/ijerph192316029_

Round 1

Reviewer 1 Report

The proportion of old sources is extremely high, and thus the research does not reflect the current picture. Try and provide more references to support your ideas that are typically substantiated by only one source – and as recent & relevant as possible. For example, ‘Adolescence is a critical time for mental wellbeing as almost 75% of adult mental illnesses have origins during childhood and youth’ is supported by only one source, 15 years old. ‘In 2019, Goodyer and Wilkinson (2019)’ – remove ‘In 2019’. ‘It has long been recognized that, with few exceptions, antidepressants are arguably not better than placebo and are not recommended for adolescents with depression (National Institute for Health and Care Excellence, 2019).’ – try and focus on recent peer-reviewed journal sources, and more than one. There are several double source repetitions. E.g., ‘In their 2019 review, Goodyer and Wilkinson concluded the most effective antidepressant was fluoxetine with some evidence for sertraline and escitalopram (Goodyer & Wilkinson, 2019).’ ‘Vitiello and Ordonez (2016) found unconvincing overall evidence for anti-depressants with only two studies showed improvement with fluoxetine compared to placebo (Cipriani et al., 2016; Ma et al., 2014)’ – poorly constructed. A Conclusions section is needed.  

Author Response

Reviewer one

In response to reviewer one’s recommendations we have responded to recommendations and included additional more recent up to date peer review articles to substantiate our ideas and to make the paper more current and relevant. The typographical changes were made as suggested by removing the double sourcing. We have attempted to improve sentence construction. A conclusion has also been added.

Author Response

Reviewer two

As with reviewer one we have included additional more recent up to date peer reviewed articles with several articles to affirm the specific ideas and arguments expressed. We have include further comments and information regarding the risk factors to support the life cycle period. The self-perception section has been expanded with more peer review articles supporting the arguments. In addition we have outlined in more detail the proposed model. The rules of the references have been adjusted appropriately. The article noted which was not cited in the text has been adjusted.

Reviewer 3 Report

Congratulations to the authors for the initiative and the importance of the chosen issue. The start is promising, but I have a few questions and observations:

The wording of the objective in the summary is confusing and incomplete. The word "Objective:" should not appear after the word "Abstract:". 

In the introduction the word "approach" appears too many times.

Current approaches to diagnosis and treatment of adolescent depression are currently  largely based upon adult depressive criteria with a confusing lack of clarity or uniformity of approach regarding the various treatment options (Cytryn & McKnew, 1972).  - This argument is based on a very old reference. Are there really no clear intervention protocols? What exactly are those fuzzy approaches and what exactly are they missing?

The paradigm proposed by the authors has in the foreground the role of self-perception and the effects of developmental disruptions on it. What is different and in addition to this paradigm compared to the CBT model or the Schema Therapy  for example?

An outline of the paradigm may facilitate understanding of the underlying principles.

Author Response

Reviewer three

The abstract has been adjusted accordingly and the removal of wording adjusted as suggested with the reduction of the use of the word ‘approach’.  Additional more recent peer review articles have been included. As requested the paradigm has been explained more fully utilising developmental concepts. How the concept interacts with schemas and CBT are explained.

Round 2

Reviewer 1 Report

This revised version can be published.